# Construction of Ovarian Cancer Prognostic Model Based on the Investigation of Ferroptosis-Related lncRNA

**DOI:** 10.3390/biom13020306

**Published:** 2023-02-06

**Authors:** Shaoyi Yang, Jie Ji, Meng Wang, Jinfu Nie, Shujie Wang

**Affiliations:** 1Anhui Province Key Laboratory of Medical Physics and Technology, Institute of Health and Medical Technology, Hefei Institutes of Physical Science, Chinese Academy of Sciences, Hefei 230031, China; 2Science Island Branch, Graduate School of University of Science and Technology of China, Hefei 230026, China

**Keywords:** ovarian cancer, ferroptosis, lncRNAs, immune cell infiltrate, prognosis

## Abstract

(1) Background: Ovarian cancer (OV) has the high mortality rate among gynecological cancers worldwide. Inefficient early diagnosis and prognostic prediction of OV leads to poor survival in most patients. OV is associated with ferroptosis, an iron-dependent form of cell death. Ferroptosis, believed to be regulated by long non-coding RNAs (lncRNAs), may have potential applications in anti-cancer treatments. In this study, we aimed to identify ferroptosis-related lncRNA signatures and develop a novel model for predicting OV prognosis. (2) Methods: We downloaded data from The Cancer Genome Atlas (TCGA), Genotype-Tissue Expression, and Gene Expression Omnibus (GEO) databases. Prognostic lncRNAs were screened by least absolute shrinkage and selection operator (LASSO)-Cox regression analysis, and a prognostic model was constructed. The model’s predictive ability was evaluated by Kaplan–Meier (KM) survival analysis and receiver operating characteristic (ROC) curves. The expression levels of these lncRNAs included in the model were examined in normal and OV cell lines using quantitative reverse transcriptase polymerase chain reaction. (3) Results: We constructed an 18 lncRNA prognostic prediction model for OV based on ferroptosis-related lncRNAs from TCGA patient samples. This model was validated using TCGA and GEO patient samples. KM analysis showed that the prognostic model was able to significantly distinguish between high- and low-risk groups, corresponding to worse and better prognoses. Based on the ROC curves, our model shows stronger prediction precision compared with other traditional clinical factors. Immune cell infiltration, immune checkpoint expression levels, and Tumor Immune Dysfunction and Exclusion analyses are also insightful for OV immunotherapy. (4) Conclusions: The prognostic model constructed in this study has potential for improving our understanding of ferroptosis-related lncRNAs and providing a new tool for prognosis and immune response prediction in patients with OV.

## 1. Introduction

According to the Global cancer statistics 2020, 207,252 patients died of ovarian cancer (OV) worldwide in 2020 [1]. OV is one of the most lethal gynecological malignancies. [2]. Because of a lack of typical signs and symptoms, most patients are diagnosed at an advanced stage, leading to a poor survival rate [3].

Currently, human epididymis protein 4 (HE4) and carbohydrate antigen 125 (CA-125) are the most common biomarkers for predicting OV but have low sensitivity and specificity [4,5]. In addition, other potential markers, including circulating tumor DNA (ctDNA) [6], microRNAs (miRNAs) [7], and long non-coding RNAs (lncRNAs) [8], have been extensively studied in recent years and provide potential new methods for early diagnosis and prognosis. Research on lncRNA-based prognostic markers for OV has led to the development of prognostic models, including immune [9], autophagy [10], and N6-methyladenosine-related lncRNA prognostic models [11]. We are currently investigating a new method to improve prognosis of patients with OV.

Ferroptosis, an iron-dependent cell death mechanism identified in 2012, is typically accompanied by the accumulation of intracellular reactive oxygen species (ROS) [12]. There is a potential for developing new anti-cancer drugs that can induce ferroptosis in tumor cells and are characterized by lower drug resistance [13]. Tumor cells undergoing ferroptosis can activate anti-tumor immune responses [14]. You et al. found that iron ion transport genes, such as *TFR1*, *FPN*, and *Tf*, were abnormally expressed in OV cells [15]. Ferroptosis was also reported to inhibit the growth of OV cells and induce anti-tumor effects [16]. Therefore, given the associations of ferroptosis with tumor immunity and tumorigenesis, analyzing this process in tumors may help researchers to develop more efficient treatment strategies.

LncRNAs are defined as RNA molecules greater than 200 nucleotides in length that are not translated into protein. However, they play significant roles in many biological regulatory processes, including ferroptosis [17]. Zheng et al. [18] identified 10 ferroptosis-related lncRNAs that could be useful in predicting lung adenocarcinoma outcomes. Tang et al. [19] identified 25 lncRNAs, then revealed a lncRNA signature, with potential for predicting head and neck squamous cell carcinoma. However, to the best of our knowledge, there have been few studies on ferroptosis-related lncRNAs in OV. Therefore, our study aimed to determine an OV prognostic signature, based on ferroptosis-related lncRNAs, that can possibly help to predict the overall survival (OS) and immunotherapy efficacy of patients with OV.

## 2. Materials and Methods

### 2.1. Data Collection

To obtain transcriptomic, clinicopathological, and survival data of patients with OV, we downloaded data for 515 samples from The Cancer Genome Atlas (TCGA) and Genotype-Tissue Expression (GTEx) databases as the training cohort. Of these, 427 tumor samples were obtained from TCGA and 88 normal samples were obtained from GTEx (Table 1). The integrated TCGA and GTEx data were downloaded from the UCSC Xena database (http://xena.ucsc.edu/, accessed on 1 November 2021). The GSE102073 dataset, containing data of 85 tumor samples with clinical information, including age, tumor stage, progression-free survival, and OS, was then selected from the Gene Expression Omnibus (GEO) database as the validation cohort (Table 1). In addition, 259 ferroptosis-related genes (Appendix A) were downloaded from FerrDb, which is an updated database that provides information on ferroptosis, such as markers, related molecules, and diseases [20].

### 2.2. Screening for Differentially Expressed Ferroptosis-Related lncRNAs

Differential analysis was performed between the 427 tumor samples and 88 normal samples in the training cohort. The data were considered statistically significant if |logFC| > 2 and adj. *p* value < 0.05 for the differentially expressed lncRNAs. Pearson correlation analysis was performed between differentially expressed lncRNAs and ferroptosis-related genes to identify differentially expressed ferroptosis-related lncRNAs (DEfrlncRNAs). The association was considered significant if the correlation coefficient was |R| > 0.3 and *p* < 0.001. Gene Ontology (GO) and the Kyoto Encyclopedia of Genes and Genomes (KEGG) enrichment analyses were used to elucidate the biological function in which ferroptosis-related genes are involved.

### 2.3. Construction and Validation of Prognostic Models

We first identified DEfrlncRNAs associated with OS using univariate Cox regression, followed by least absolute shrinkage and selection operator (LASSO) regression analysis for further screening. Multivariate Cox analysis was then used to further select the independent DEfrlncRNAs associated with prognosis to construct a prognostic model based on the training cohort (FL_Score, ferroptosis-related lncRNA score). The specific calculation formula is as follows: FL_Score = (β1_(lncRNA1)_ × expression of lncRNA1) + (β2_(lncRNA2)_ × expression of lncRNA2) + … + (βn_(lncRNAn)_ × expression of lncRNAn), where β is the regression coefficient of each lncRNA and expression is the expression level of the corresponding lncRNA. The cutoff value of the FL_Score was calculated using the surv_cutpoint function of the survminer R package. The results were divided into two groups: low-risk group (FL_Score less than the cutoff value) and high-risk group (FL_Score greater than or equal to the cutoff value). The predictive accuracy of the model was assessed using time-dependent receiver operating characteristic (ROC) curves in the training and validation cohorts.

### 2.4. Construction of a Predictive Nomogram

Clinicopathological factors were combined with our constructed risk scores to construct a nomogram to predict the prognosis of patients with OV.

### 2.5. Analysis of Immune Cell Infiltration in the Tumor Microenvironment (TME)

We quantified and evaluated the relative abundance of different immune cell types in the low-risk and high-risk groups using the CIBERSORT algorithm to compare and predict immune cell infiltration between the two groups.

### 2.6. Gene Set Enrichment Analysis (GSEA)

GSEA is an enrichment analysis method based on gene sets [21]. We used GSEA to compare biological pathway differences between the two groups using the clusterProfiler package. The predefined gene set was obtained from the MSigDB database [22]. A false discovery rate (FDR) < 0.05 was considered significant.

### 2.7. Cell Culture

Human normal ovarian epithelial cell line (IOSE80 cell line) and ovarian cancer cell lines (SKOV-3 and A2780 cell lines) were purchased from Hunan Fenghui Biotechnology Co., Ltd. (Hunan, China). Cells were cultured in DMEM (Gibco, Billings, MT, USA), 10% fetal bovine serum (Gibco), 100 U/mL penicillin, and 100 U/mL streptomycin in an incubator at 37 °C and 5% CO_2_ concentration.

### 2.8. RNA Isolation and Quantitative Reverse Transcriptase Polymerase Chain Reaction (qRT-PCR)

We used TRIzol^®^ Plus RNA Purification Kit (Invitrogen, Waltham, MA, USA, product number: 12183-555) and BCP phase separation reagents (Hete Biotechnology Co., Timonium, MD, USA, product number: RR009) to isolate RNA in a high-speed freezing centrifuge (Cence). We used the spectrophotometer (Implen, München, Germany, NanoPhotometer-NP80) to measure RNA concentration and stored samples at −80 °C. Reverse transcription was performed using PrimeScript™ RT Master Mix (Perfect Real Time) (TAKARA, Kusatsu, Japan, product number: RR036A) in a dry bath (DLAB, product number: HB120-S). Quantitative reverse transcriptase polymerase chain reaction (qRT-PCR) was performed using TB Green Premix Ex Taq II (Tli RNaseH Plus) (TAKARA, product number: RR820A) on StepOnePlus Real-Time PCR System (ABI). Appendix A lists all the primer sequences.

### 2.9. Data Analysis

We used the Bioconductor package in R software (version 4.1.0) to perform data analysis. Differential analysis was carried out using the limma package, and heatmap plotting was performed using the pheatmap package. Kaplan–Meier (KM) survival analysis and univariate and multivariate Cox regression analyses were performed using the survival package. LASSO regression analysis was carried out using the glmnet package. Time-dependent ROC curves were generated using the timeROC package. Nomograms were plotted using the regplot package. Violin plots were generated using the VioPlot package. Immune cell infiltration analysis was carried out using the CIBERSORT algorithm [23,24]. The input mixture matrix is our gene expression matrix, and the input of the gene signature for 22 immune cell types is from Newman et al. [23]. Prediction of immune checkpoint blockade (ICB) response was performed using the Tumor Immune Dysfunction and Exclusion (TIDE) tool [25,26].

## 3. Results

### 3.1. Differential Analysis and Enrichment Analysis of Ferroptosis-Related Genes

In total, we identified 113 differentially expressed ferroptosis-related genes (63 upregulated and 50 downregulated in tumors) (Figure 1). The GO results (Figure 2A) suggested that the differentially expressed genes are mainly involved in biological processes (BP) associated with response to oxygen stress, cellular response to chemical stress, and response to nutrient levels. In the cellular component (CC), they are related to the apical part of the cell and apical plasma membrane, whereas the molecular function (MF) mainly regulates oxidoreductase activity and iron ion binding. The results of the KEGG analysis show that the differentially expressed genes are mainly involved in processes concerning chemical carcinogenesis-reactive oxygen species, lipids, and atherosclerosis (Figure 2B).

### 3.2. Ferroptosis-Related lncRNAs-Based Prognostic Signature

Differential analysis was performed between 427 tumor samples and 88 normal samples, and 12,505 differentially expressed lncRNAs were identified. Pearson correlation analysis was then performed between differentially expressed lncRNAs and ferroptosis-related genes, where 1748 DEfrlncRNAs were identified based on the criteria of correlation coefficient |R| > 0.3 and *p* < 0.05. A total of 157 prognosis-associated DEfrlncRNAs were detected by univariate Cox regression and 44 additional lncRNAs were identified using LASSO regression analysis. Finally, 18 independent prognosis-associated DEfrlncRNAs were detected using multivariate Cox regression analysis (AC007796.1, TLR8-AS1, RP11-713M15.2, CTB-171A8.1, LBX2-AS1, CTD-2130F23.2, RP11-88G17.6, RP11-388M20.1, RP11-678G14.3, RP4-650F12.2, RP11-701H24.7, RP11-1018N14.5, LINC01281, RP11-301G19.1, CTD-2330K9.3, AP000344.3, CTD-2506J14.1, and AC078842.3) and used to construct the prognostic model. An expression heatmap of these 18 lncRNAs was plotted (Figure 3A). Using Multi-Experiment Matrix (MEM) analysis [27], we predicted the target mRNAs of the 18 lncRNAs and the results were visualized using Cytoscape software version 3.9.1 (Figure 3B). GO enrichment analyses were used to investigate the biological process (BP), cellular component (CC), and molecular function (MF) in which the target mRNAs are involved. The results indicated these mRNAs are mainly involved in metal ion transmembrane transporter activity and channel activity, which may be associated with the regulation of iron ion transporting (Figure 3C).

### 3.3. Model Construction and Validation

The FL_Score of each patient was calculated based on the constructed prognostic model, and patients were divided into high- and low-risk groups according to the FL_Score. The KM survival analysis indicated that there was a significant difference (*p* < 0.0001) in survival rates between the high- and low-risk groups (Figure 4A). A higher FL_Score corresponded to poorer survival. We found that the FL_Score and survival rates of patients with OV were negatively correlated (Figure 4F). The ROC curves of the prognostic model (Figure 4C) showed area under the ROC curve (AUC) values of 0.762, 0.793, and 0.785 for the prediction of 1-year, 3-year, and 5-year survival rates, respectively. These data indicate that this model has high potential for predicting the prognosis of OV. The ROC curves of the different clinical factors and FL_Score to predict the 3-year survival rate (Figure 4B) showed that the AUC values of FL_Score, age, grade, and stage were 0.793, 0.623, 0.5, and 0.545, respectively, indicating that the FL_Score had a better ability to predict the prognosis of OV when compared with the other clinical factors.

To validate the potential of the FL_Score to predict prognosis, we calculated the FL_Score based on the validation dataset GSE102073. From the KM survival analysis results, there was an apparent significant difference in the survival rates between the high- and low-risk groups (*p* = 0.013) (Figure 4D). Furthermore, a higher FL_Score corresponded to poorer survival. The ROC curves of different clinical factors and the FL_Score for predicting the 5-year survival rate (Figure 4E) showed that the AUC values of the FL_Score, age, and stage were 0.681, 0.578, and 0.65, respectively, indicating that FL_Score has a better capability to predict OV prognosis when compared with the other clinical factors.

We also compared the 1-, 3-, and 5-year AUC values of FL_Score with 18 other published prognostic signatures [9,10,28,29,30,31,32,33,34,35,36,37,38,39,40,41,42] for patients with OV in TCGA database to test the prognostic performance of our FL_Score signature (Figure 5). These 18 signatures exhibited associations with some biological features, including N6-methyladenosine, pyroptosis, autophagy, and others. Our FL_Score signature displayed better performance than any other signatures in the TCGA-OV cohort. Forest plots of univariate Cox (hazard ratio (HR): 2.72, 95% confidence interval (CI): 2.32–3.19, *p* < 0.001) (Figure 6A) and multivariate Cox (HR: 2.72, 95% CI: 2.31–3.19, *p* < 0.001) (Figure 6B) analyses revealed that FL_Score was a prognostic predictor independent of grade and stage.

### 3.4. Construction of a Nomogram for Predicting Prognosis

We also generated a nomogram that combines the clinical factors and a ferroptosis-related lncRNA prognostic signature (Figure 6C). The nomogram includes grade and stage, which are important clinical factors in OV prognosis prediction, along with the FL_Score. Each item was evaluated according to the actual condition of the patient and then combined to obtain the total score of the patient, which was then compared with the nomogram to predict the patient’s 1-, 3-, and 5-year survival rates.

### 3.5. Gene Set Enrichment Analysis (GSEA)

The GSEA results (Figure 7) suggested that immune and cell cycle related pathways, such as the T cell receptor signaling pathway, primary immunodeficiency, natural killer cell-mediated cytotoxicity, and the cell cycle were enriched in the low-risk group.

### 3.6. Correlation between Immune Cell Infiltration and FL Score

Based on the gene expression matrix, we used the CIBERSORT algorithm to predict the immune cell infiltration of the tumor samples. We found that the infiltration score of naive B cells and the FL_Score were positively correlated (*p* < 0.05), meaning that there were more naive B cells infiltrating high-risk tumors. Conversely, the infiltration score of M1 macrophages (*p* < 0.001) and resting dendritic cells (*p* < 0.05) were negatively correlated with the FL_Score, indicating that fewer M1 macrophages and dendritic cells infiltrated the high-risk tumors (Figure 8).

### 3.7. Immune Checkpoint Expression Levels and ICB Response Prediction

Because of the importance of immune checkpoint inhibitors in immunotherapy, the differences in mRNA expression levels of immune checkpoint between the high- and low-risk groups indicate that there are significant differences in the expression levels of molecules, such as CTLA4, BTLA, CD200R1, CD200, and TNFRSF4, between the two groups of patients (Figure 9A).

Our results indicate that patients in the high-risk group had significantly higher TIDE scores than those in the low-risk group (Figure 9B). Higher TIDE scores were associated with poorer ICB responses, which were specifically associated with poorer survival in patients treated with ICB therapy.

### 3.8. LncRNA Expression Levels in Cell Lines

The ferroptosis-related lncRNA expression profiles from the TCGA-OV dataset indicate that most of the lncRNAs included in the model were upregulated in OV tissues, including CTB-171A8.1, AC078842.3, TLR8-AS1, RP11-1018N14.5, RP11-713M15.2, RP11-301G19.1, RP11-88G17.6, LINC01281, LBX2-AS1, CTD-2506J14.1, and CTD-2330K9.3 (Figure 10). To examine the expression levels of these lncRNAs in vitro, qRT-PCR experiments were performed using two OV cell lines (SKOV-3 and A2780) and one normal ovarian epithelial cell line (IOSE80). In vitro results showed that CTB-171A8.1, AC078842.3, TLR8-AS1, RP11-1018N14.5, RP11-713M15.2, RP11-301G19.1, RP11-88G17.6, LINC01281, LBX2-AS1, CTD-2506J14.1, and CTD-2330K9.3 were significantly upregulated in SKOV-3 and A2780 cells compared with IOSE80 cells. These results were consistent with, and validated, our database analysis findings (Figure 11).

## 4. Discussion

There is growing evidence that ferroptosis plays an important role in numerous diseases, including various cancers [43]. In the future, ferroptosis may be involved in the development of novel cancer treatment methods [44,45]. Killing cancer cells by inducing apoptosis is one of the most common approaches for cancer treatment. However, in recent years, cancer cells have been found to be drug resistant and often insensitive to apoptosis-inducing chemotherapeutic drugs [46]. Therefore, many recent studies have focused on utilizing ferroptosis and other regulatory cell death mechanisms as treatment options. Sorafenib is a drug used to treat advanced hepatocellular and pancreatic cancers and works by inhibiting system X_c_- and MT-1G in the ferroptosis system [47,48]. Artemisinin increases intracellular levels of free iron ions and thus also participates in the regulation of ferroptosis [49,50].

The specific mechanisms regulating ferroptosis remain unclear, but miRNAs and lncRNAs are being increasingly recognized as key mediators of such regulation [44]. The interaction between lncRNA P53RRA and GTPase activating protein binding protein 1 (G3BP1) in the cytoplasm causes p53 to remain in the nucleus, leading to ferroptosis [51]. The gene *LINC00336* is upregulated and acts as a competing endogenous RNA (ceRNA) oncogene in lung cancer, which suppresses ferritin formation [52]. Undoubtedly, lncRNAs are extensively involved in the large and complex regulatory network of ferroptosis, making it essential to conduct studies targeting various tumors based on ferroptosis-related lncRNAs.

In this study, we first identified ferroptosis-related genes that are differentially expressed between tumor and normal samples, then performed GO and KEGG enrichment analyses. We found that the tumorigenesis- and development-related pathways were significantly enriched, including the HIF-1 and p53 signaling pathways. The HIF-1 signaling pathway plays a pivotal role in tumor proliferation and migration [53]. HIF-1 coordinates the activities of signaling molecules that affect tumorigenesis by cooperating with many other factors, including STAT3, NF-κB, Notch, and other pathways [54]. In OV, HIF-1α binds to the AEG-1 promoter and induces its transcriptional upregulation. Increased AEG-1 expression is associated with OV metastasis [55]. In addition, knocking down HIF-1α expression could improve the response of OV cells with cisplatin resistance. HIF-1α is expected to be a new target for such treatments, potentially improving the efficacy of cisplatin treatment [56]. The *p53* gene is one of the most studied and important tumor suppressors, with approximately half of all tumors harboring *p53* mutations or deletions [57]. In colorectal cancer, TP53 inhibits ferroptosis in cancer cells by promoting DPP4 localization in the nucleus of non-enzymatically active cells, thereby promoting cancer cell growth [58].

Here, we identified 18 ferroptosis-related lncRNAs as risk measures. Among them, lncRNA TLR8-AS1 has been identified as a cancer-related fibroblast regulatory lncRNA that enhances cell metastasis and chemoresistance in OV in vitro and in vivo. It can also promote OV metastasis and chemoresistance by activating the NF-κB pathway [59]. This lncRNA has the potential to be used as a therapeutic target or diagnostic indicator. The high expression levels of LBX2-AS1 were associated with OV progression, whereby it promotes tumor cell proliferation and migration [60,61]. In addition, LBX2-AS1 has been shown to be associated with the development of various tumors, including gastric cancer [62], colorectal cancer [63], and glioma [64]. RP11-301G19.1 regulates the proliferation and apoptosis of multiple myeloma cancer cells by targeting the miR-582-5p/HMGB2 axis [65], but to our knowledge, its role in OV development has not yet been reported. Similarly, AP000344.3 is reportedly associated with the development of bladder cancer [66]; however, few studies have reported its role in OV development. Apart from the abovementioned lncRNAs, the additional 14 lncRNAs (AC007796.1, RP11-713M15.2, CTB-171A8.1, CTD-2130F23.2, RP11-88G17.6, RP11-388M20.1, RP11-678G14.3, RP4-650F12.2, RP11-701H24.7, RP11-1018N14.5, LINC01281, CTD-2330K9.3, CTD-2506J14.1, and AC078842.3) have not yet been reported in cancer studies. The expression profile of the 18 lncRNAs in the TCGA-OV dataset indicated that CTB-171A8.1, AC078842.3, TLR8-AS1, RP11-1018N14.5, RP11-713M15.2, RP11-301G19.1, RP11-88G17.6, LINC01281, LBX2-AS1, CTD-2506J14.1, and CTD-2330K9.3 were upregulated in OV tissues. We examined the expression levels of these lncRNAs in a normal ovarian epithelial cell line (IOSE80) and two OV cell lines (SKOV-3 and A2780). The results showed that these 11 lncRNAs were overexpressed in all OV cell lines, indicating that they may be involved in promoting OV progression.

We constructed an OV prognostic model based on the DEfrlncRNAs, which demonstrated good OV prognosis predicting ability using ROC curves. In addition, we combined two common clinical factors, grade and stage, with our prognostic model and constructed a nomogram to provide a visual and quantitative method for predicting the 1-, 3-, and 5-year survival of patients with OV.

We performed an immune cell infiltration analysis using CIBERSORT and examined the correlation between immune cell infiltration and FL_Score. Macrophages are immune cells that are widely distributed throughout the body with high heterogeneity. As a typical phagocyte of the monocyte macrophage system, macrophages are involved in non-specific and specific immune regulation by phagocytosing bacteria, dead cells, and cellular debris. Among them, M1 macrophages are a type of classically activated macrophage that can kill tumor cells. M1 macrophages can promote or initiate an anti-tumor immune process by producing strong oxidative free radicals, secreting multiple immunogenic cytokines, and presenting tumor antigens to T cells [67]. Classically activated macrophages (M1 macrophages) can kill a wide range of tumor cells, including breast cancer cells [68]. In our study, we found that patients with a higher FL_Score had lower M1 macrophage infiltration scores, indicating that the tumors in high-risk patients may have fewer infiltrating M1 macrophages.

B cells are pluripotent stem cells derived from the bone marrow that can differentiate into plasma cells when stimulated by antigens. Plasma cells can synthesize and secrete antibodies for humoral immunity. Many studies have reported that B cells play important roles in promoting tumor development and affecting patient prognosis. For example, the presence of B cells in the TME can promote the invasion and metastasis of bladder cancer [69], reduce the survival rate of patients with kidney cancer [70,71], and increase the risk of disease recurrence after prostatectomy [72]. Furthermore, patients with tumors that show more plasma cell and CD138^+^ B cells infiltration have shorter recurrence-free survival [73]. Similar findings were reported in mouse models. For example, B-cell-deficient mice have more T cells present in their tumors and suppress the anti-tumor CTL and T(H)1 cytokine responses [74]. Experiments on mouse models of melanoma, B-cell lymphoma, and sarcoma reveal that B-cell and B-cell-derived cytokine IL-10 can suppress anti-tumor immune responses [75]. In our study, we found that patients with a higher FL_Score had higher naive B-cell infiltration scores, indicating that the tumors in high-risk patients may be infiltrated by more naive B cells. In summary, we found that high-risk tumors (higher FL_Scores) are infiltrated by more tumor-promoting immune cells and fewer tumor-suppressing immune cells. Analyzing the relationship between tumor risk and immune cell infiltration provides strong evidence supporting our newly developed OV prognostic model.

TIDE is a computational method used to predict ICB response [25,26]. According to the TIDE prediction results, patients in the high-risk group showed a worse response to ICB than those in the low-risk group. In addition, immune checkpoints with significantly different expression levels between the high- and low-risk groups are useful for guiding OV immunotherapy, but further experimental validation of the specific therapeutic effect is needed.

Overall, the prognostic signature based on ferroptosis-related lncRNAs provides new insights for potential ferroptosis-based clinical treatment and survival prediction methods. We also performed immune cell infiltration and immune checkpoint expression level analyses to identify the association between FL_Score and immunotherapy response.

## 5. Conclusions

In conclusion, we constructed a ferroptosis-related lncRNA-based prognostic model for OV and validated it using patient data. We identified a signature that included 18 lncRNAs and constructed a risk score model that showed a potential application for predicting the survival and immune response of patients with OV. We believe that this work may provide a new tool for prognosis and immune response prediction for these patients.

## Figures and Tables

**Figure 1 biomolecules-13-00306-f001:**
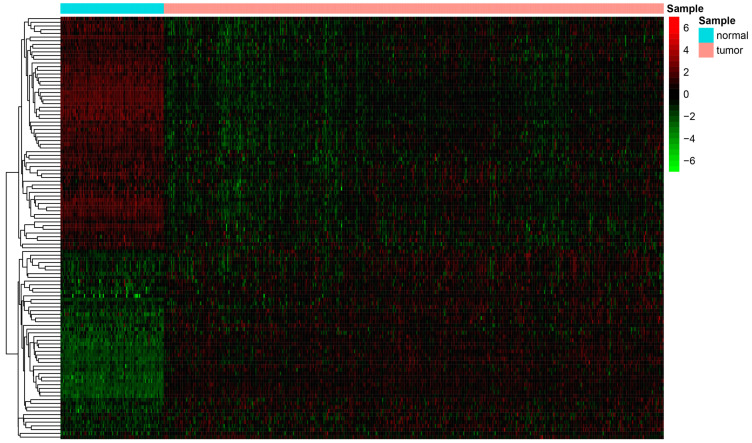
Heatmap depicting differentially expressed ferroptosis-related genes.

**Figure 2 biomolecules-13-00306-f002:**
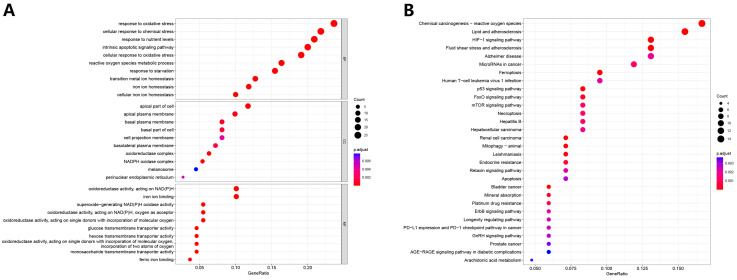
Gene ontology (GO) and Kyoto Encyclopedia of Genes and Genomes (KEGG) enrichment analyses of differentially expressed ferroptosis-related genes. (**A**) GO results. (**B**) KEGG results.

**Figure 3 biomolecules-13-00306-f003:**
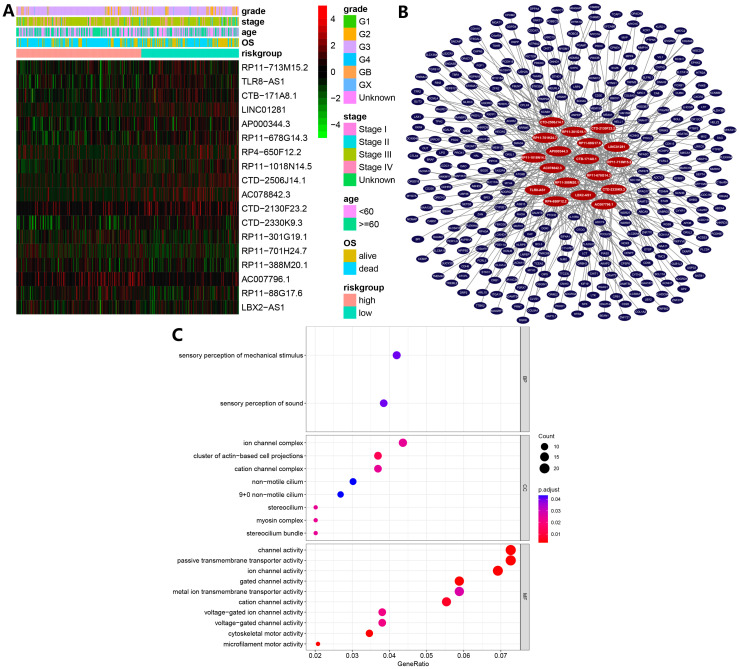
(**A**) Heatmap of ferroptosis-related long non-coding RNA (lncRNA) prognosis signature and clinicopathological manifestations. (**B**) Target mRNAs of the 18 lncRNAs predicted by Multi-Experiment Matrix (MEM) analysis. (**C**) Gene ontology (GO) enrichment analysis of target mRNAs of the 18 lncRNAs predicted.

**Figure 4 biomolecules-13-00306-f004:**
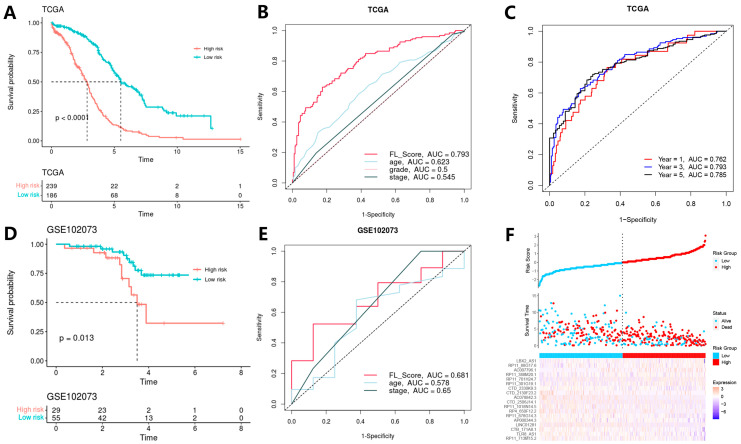
Results of model validation in the training and validation cohorts. (**A**) Kaplan–Meier (KM) survival analysis in the training cohort. (**B**) Receiver operating characteristic (ROC) curves for 3-year survival. (**C**) ROC curves for ferroptosis-related lncRNA score (FL_Score). (**D**) KM survival analysis in the validation cohort. (**E**) ROC curves for 5-year survival. (**F**) Risk survival status plot for the training cohort.

**Figure 5 biomolecules-13-00306-f005:**
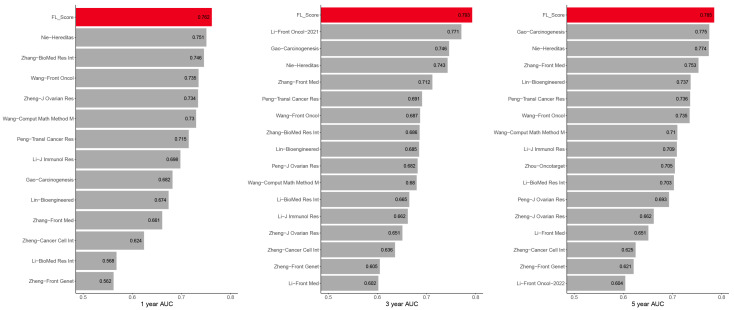
Comparisons between the ferroptosis-related lncRNA score (FL_score) signature and other models [9,10,28,29,30,31,32,33,34,35,36,37,38,39,40,41,42]. The 1-, 3-, and 5-year area under the curve (AUC) values of the FL_score signature (red column) and other models (gray column) developed in The Cancer Genome Atlas (TCGA) ovarian cancer cohort.

**Figure 6 biomolecules-13-00306-f006:**
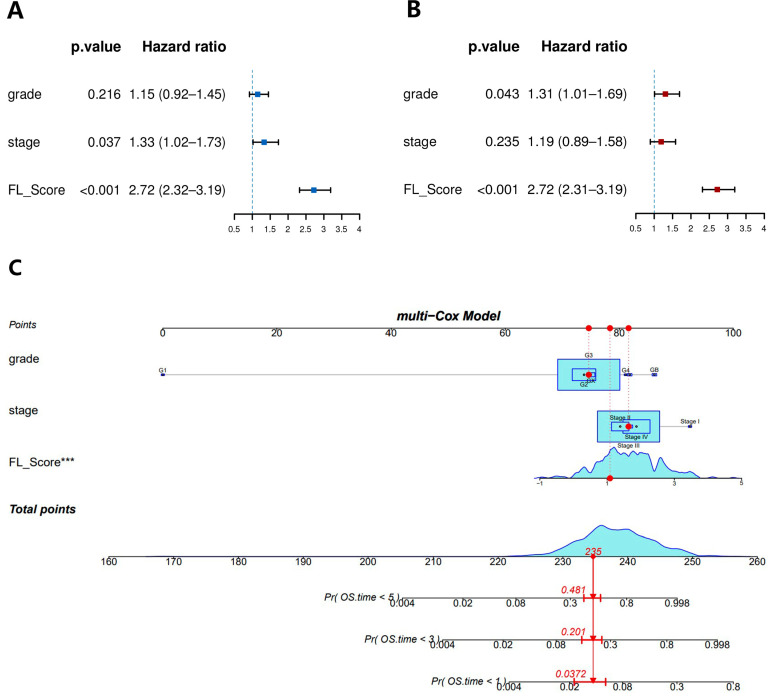
(**A**) Forest plot of univariate COX analysis. (**B**) Forest plot of multivariate COX analysis. (**C**) A nomogram for both clinicopathological factors and ferroptosis-related lncRNA score (FL_score). *** (*p* < 0.001).

**Figure 7 biomolecules-13-00306-f007:**
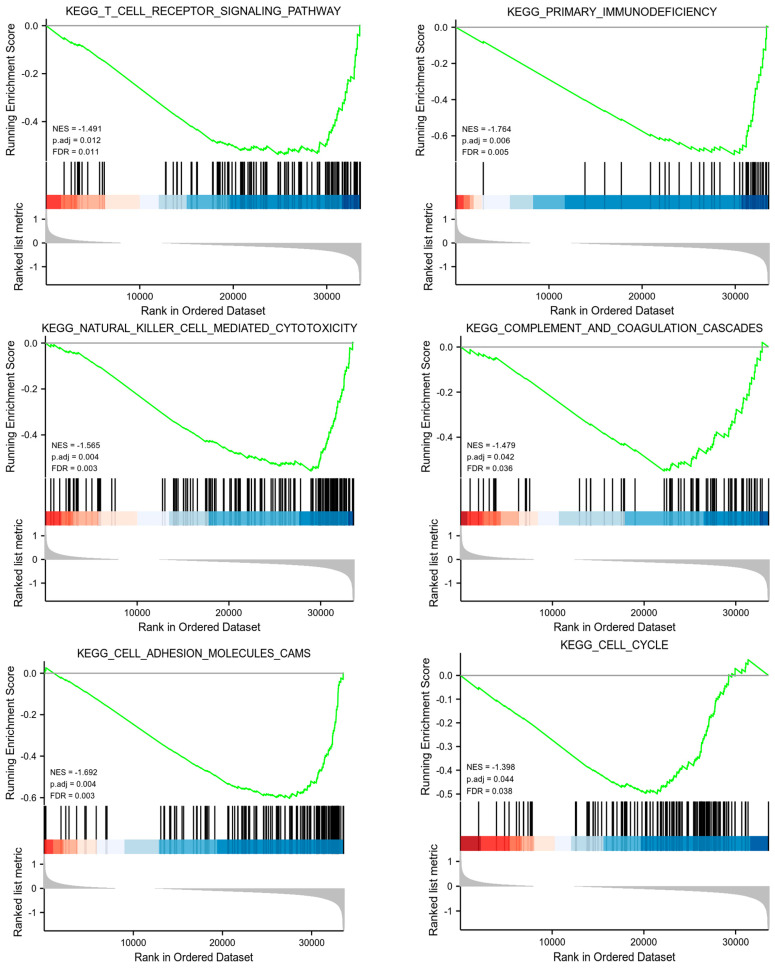
Gene set enrichment analysis (GSEA) for model long non-coding RNAs (lncRNAs). NES, normalized enrichment score; FDR, false discovery rate.

**Figure 8 biomolecules-13-00306-f008:**
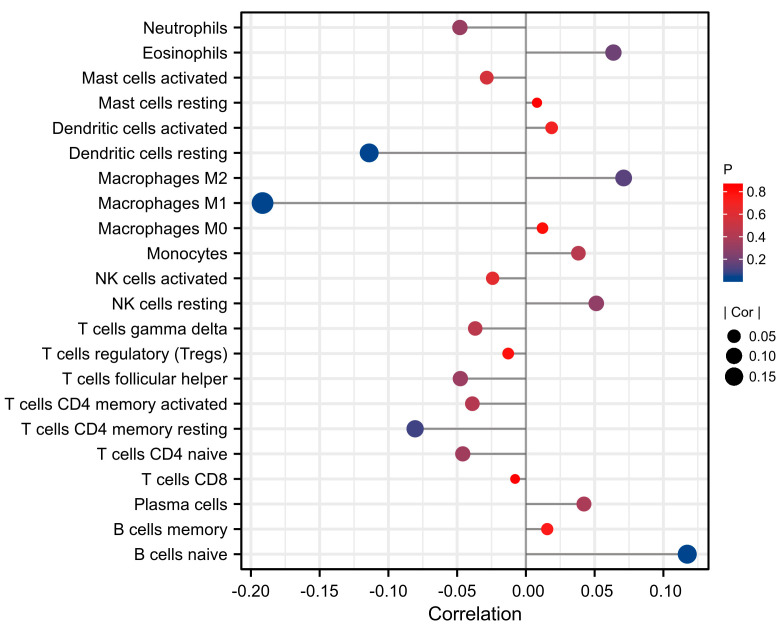
The relationship between immune cell infiltration analysis using the CIBERSORT algorithm and ferroptosis-related lncRNA score (FL_score).

**Figure 9 biomolecules-13-00306-f009:**
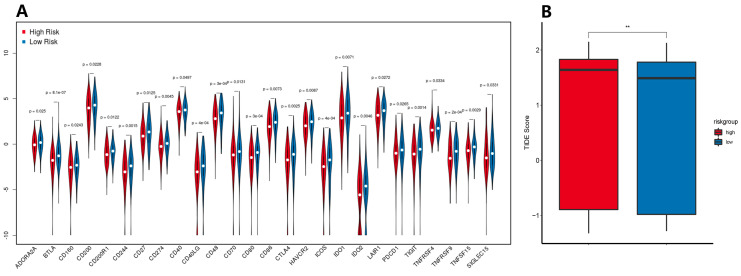
(**A**) Expression levels of immune checkpoint genes in the low- and high-risk groups. (**B**) Tumor Immune Dysfunction and Exclusion (TIDE) score in the low- and high-risk groups. ** *p* < 0.01.

**Figure 10 biomolecules-13-00306-f010:**
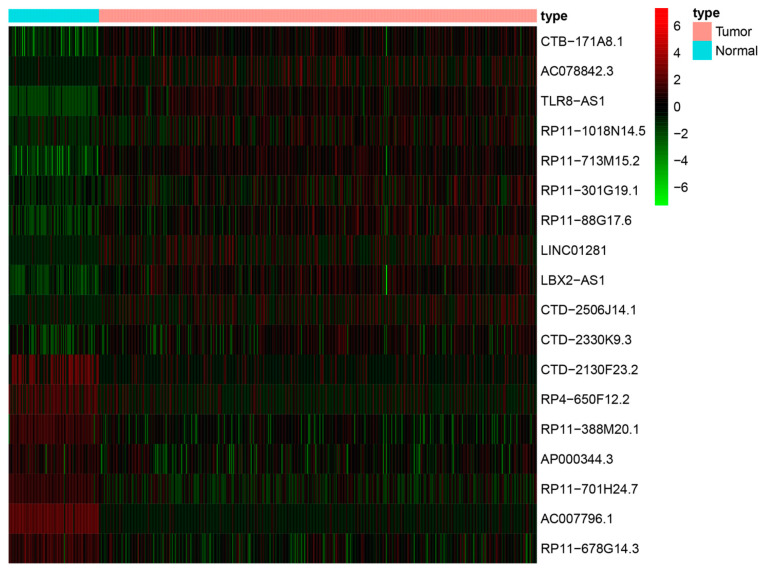
Expression profile of ferroptosis-related long non-coding RNAs (lncRNAs) from The Cancer Genome Atlas (TCGA) ovarian cancer dataset included in the model.

**Figure 11 biomolecules-13-00306-f011:**
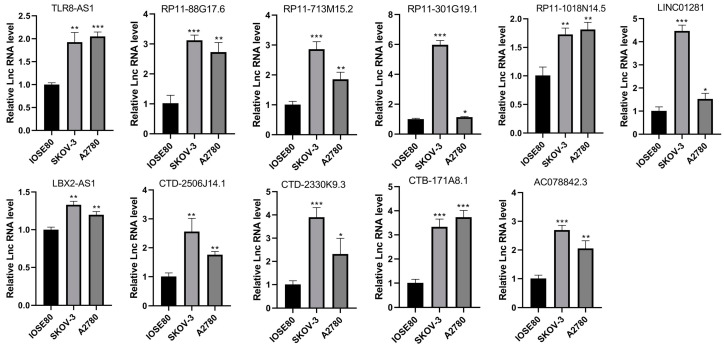
Expression levels of long non-coding RNAs (lncRNAs) in normal and ovarian cancer cell lines using qRT-PCR experiments. * *p* < 0.05; ** *p* < 0.01; *** *p* < 0.001.

**Table 1 biomolecules-13-00306-t001:** Clinical data of OV patients.

		TCGA		GSE102073	
OV Samples		427		85	
Age, mean/SD		59.60	11.41	59	8.81
Stage, *n*/%	I	1	0.23	1	1.18
	II	26	6.09	3	3.53
	III	334	78.22	56	65.88
	IV	63	14.75	25	29.41
Grade, *n*/%	1	1	0.23	NA	NA
	2	52	12.18	NA	NA
	3	363	85.01	NA	NA
	4	1	0.23	NA	NA

## Data Availability

Publicly available datasets were analyzed in this study. These data can be found here: https://www.ncbi.nlm.nih.gov/geo/ (accessed on 18 November 2021, GSE102073), http://xena.ucsc.edu/ (accessed on 1 November 2021, TCGA and GTEx).

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
