# Peer review of "Construction of Ovarian Cancer Prognostic Model Based on the Investigation of Ferroptosis-Related lncRNA"

_biomolecules, 2023, doi:10.3390/biom13020306_

Round 1

Reviewer 1 Report

The authors utilized public databases (TCGA, GTEx and GEO) to construct an 18-lncRNA prognostic model for ovarian cancer. They also validated the model using different datasets. 

Major issues

1> please detail how you integrated datasets from different databases.

2> Based on the paper (Evaluating cell lines as tumour models by comparison of genomic profiles) published in 2013, Kuramochi is a cell line most similar to HGSOC. It will be better if you can validate your results using this cell line.

3> When you used CIBERSORT to infer immune cell abundance scores, what kind of data you used for input to CIBERSORT ?

Minor issues

1> In figure3, what do GB and GX mean in grade category ?

Reviewer 2 Report

In this paper, the author constructed a ferroptosis-related lncRNA-based prognostic model for OV and validated it in patients with OV, which may provide a new tool for prognosis and immune response prediction for patients with OV. However, there are still some concerns in the following:

1.      In the “Introduction” section, most of the biomarker examples listed by the author are about the OV diagnosis. However, this paper focuses on the OV prognosis, which should be investigated and summarized more.

2.      Lots of study have identified prognosis signatures and prediction models for OV, it’s better to add comparison of 18-lncRNAs’ model with these previous published prognosis models to demonstrate the accuracy of thee 18-lncRNAs’ model.

3.      The analysis results of GSEA were abrupt in the paper, and the analysis methods were not described in detail, and it was hard to find which genes were analyzed by GSEA.

4.      Whether the 18 lncRNAs identified in this paper function through targeted mRNA? If so, the target mRNA of 18 lncRNAs can be predicted and further analysis can be conducted.

Round 2

Reviewer 2 Report

1. As for Point 2, the author showed the comparison with 18 other published prognostic signatures in fig5, but why did not 19 AUC values of 1-(A), 3-(B) and 5-(C) be displayed on each graph?

2. As for Point 4, not only the target mRNA of lncRNA should be predicted, but also the function of the target mRNA should be further analyzed.

3. The whole English of the article needs to be polished. For example, the sentences of line130 are not smooth.
